# Understanding Physicians’ Motivation to Provide Healthcare Service Online in the Digital Age

**DOI:** 10.3390/ijerph192215135

**Published:** 2022-11-16

**Authors:** Tingting Zhang, Qin Chen, William Yu Chung Wang, Yuhan Wei

**Affiliations:** 1School of Economics and Management, University of Science and Technology Beijing, Beijing 100083, China; 2School of Economics and Management, Northwest University, Xi’an 710127, China; 3Waikato Management School, University of Waikato, Hamilton 3216, New Zealand

**Keywords:** online healthcare service, physicians’ motivation, users and gratifications theory, self-determination theory

## Abstract

This study aims to investigate the factors that affect physicians’ healthcare service provision behavior on healthcare service platforms. A research model was proposed based on the related literature and uses and gratifications theory and self-determination theory. The empirical data were collected from a popular Chinese healthcare service platform, and negative binomial regression was employed to test the proposed research model. The results indicate that competence satisfaction, autonomy satisfaction, and economic benefit have positive impacts on their service provision behavior and that when physicians have a higher level of offline status, they would be less likely to provide consultation service online if they have a higher level of competence satisfaction. This study contributes to the existing literature by integrating intrinsic and extrinsic motivations to investigate how they affect physicians’ healthcare service provision behavior online. Findings from this study may derive recommendations for improving the features and design of healthcare service platforms.

## 1. Introduction

There exists a significant contradiction between the increasing demand for healthcare services and insufficient supply worldwide [1]. The World Health Organization advocates using information and communication technologies to develop telemedicine and increase access to care and medical information to improve patient outcomes. Online healthcare service platforms, such as Doctor On Demand, Crowd Med, Teladoc, and MeMD, provide a convenient channel to enhance communication between physicians and patients without the constraining of time and space, on which physicians can deliver consultation service, knowledge, and information for patients to understand disease and treatment [2]. Evidence suggests that online healthcare services have helped reduce urban-rural health disparities in both developed countries and developing countries [3,4]. Thus, online healthcare service platforms can be considered as an alternative and complementary means to the traditional healthcare service system [5]. On the one hand, online healthcare service platforms replace their traditional counterpart for delivering healthcare services, such as chronic diseases [6]. On the other hand, online healthcare service platforms complement the traditional healthcare service system by providing some services remotely, such as following up after treatment offline and patient education [7]. Especially due to the outbreak of COVID in 2019, online healthcare services have been accelerated. Online healthcare service platforms have gained a lot of attention and application and served as an important channel for healthcare delivery during the pandemic [8]. By June 2022, the total number of participants in online healthcare in China will exceed 0.32 billion, accounting for 28.5% of the country’s total number of internet users [9]. Therefore, it is important to maintain the sustainable development of such platforms.

Given the number of patients using online healthcare platforms keeps growing, it is important for these platforms to reach a critical mass of physicians who can actively make contributions because physicians are scarce healthcare resources [2]. Existing research on incentives for physicians to provide online healthcare services shows that some motivators are physicians’ individual characteristics, reputation, income, etc. [10]. It should be noted that research on physicians’ offline healthcare service behavior suggests that other professional characteristics and needs of physicians, such as service autonomy, sense of work achievement, self-development, altruism, and availability of energy and time, will also affect their behavior by influencing their individual psychology or motivation, which should also apply to online healthcare services. However, these factors and their mechanisms are not systematically reflected in the existing research on physicians’ online service behavior.

Therefore, this study aims to investigate physicians’ motivation to contribute to online healthcare service platforms. The findings of this study could not only enrich the literature on user participation on this type of platform from the service provider’s perspective but also have a big potential for recommendations on platform features and platform design. The rest of this paper is organized as follows: the next section elaborates on the related literature and the development of hypotheses, followed by descriptions of the data collection and data analysis methods in Section 3. The results are presented in Section 4, followed by discussions of the findings and conclusions in Section 5.

## 2. Literature Review and Hypothesis Development

### 2.1. Physicians’ Participation in Online Healthcare Service

Physicians are the providers of online healthcare services [11]. The willingness and behavior of physicians to provide healthcare services is an important guarantee for the supply of online healthcare services, which helps to combine online healthcare resources with offline healthcare systems [12,13]. While the willingness of physicians to provide healthcare services is a major factor that causes differences in healthcare services [14], the service behavior of physicians is not always equivalent to their willingness to provide services. Some studies have found that, even though physicians provide services on the online healthcare service platform, they are less willing to take the initiative to learn about patients, resulting in mixed service quality. Thus, although there are many studies on physicians’ intention to provide online healthcare service, it is important to investigate the motivation underlying their actual participation behavior.

Some prior studies argue that physicians’ behaviors are mostly independent of payment and driven by professional standards of care [10]. In addition, service providers’ motivation to participate in a remote mode in a nonmonetary situation is a predictor of their behavior [15]. However, other studies have found that financial rewards (such as service fees and gifts) they obtain online also promote physicians’ online service behavior [16]. In some cases, financial rewards have a greater incentive effect on physicians’ online contributions than recognition by the patients (such as the letters of thanks written by patients and online evaluations provided by patients) [1,2]. Therefore, it is necessary to further examine the role of economic incentives in motivating physicians’ actual online healthcare service provision behavior.

### 2.2. Theoretical Foundations

As an Internet-mediated platform, online healthcare platforms demonstrate a media format that requires extensive interaction among users [17]. Originating from the effectiveness perspective on media communication, uses, and gratifications (U&G) theory assumes a user-directed nature of media and that the media requires a high level of interactivity from its users [18]. This theory has been applied to examine motivational and behavioral dimensions of psychological gratification related to various technology-mediated communication modes. Thus, U&G seems appropriate for investigating service providers’ perceptions about healthcare service provision and its impact on their behaviors [19].

U&G asserts that users actively seek out the media to satisfy individual needs [20]. Therefore, they tend to be motivated to select the medium that can gratify their needs which are unable to be fully satisfied via another channel [21]. The gratification of these needs is a vital antecedent of continued use of a medium, which refers to the physician’s continuous service provision in this study. Physicians are registered in offline hospitals. They concurrently choose online platforms for the delivery of healthcare services [22]. They choose to deliver service on healthcare service platforms probably because they seek to satisfy their specific needs.

U&G clusters were resulting basic need gratifications in extrinsic and intrinsic motivations, which is explained by self-determination theory (SDT) [23]. SDT is a motivation theory which posits that motivations can be distinguished between intrinsic motivation and extrinsic motivation [24]. The former is related to the inner self, which means to do something for self-satisfaction, i.e., enjoyment, passion, competence, and relatedness. The latter is related to outer rewards, i.e., income, unemployment, and economic growth [25]. A physician’s healthcare service behavior is a result of a combination of his/her intrinsic and extrinsic motivations. Hence, this study uses the U&G theory and SDT to explain the relationship between physicians’ healthcare service provision and its motivational factors.

### 2.3. The Motivation of Healthcare Service Provision

According to SDT, people have some basic psychological needs, and their fulfillment is positively associated with high levels of self-determined motivation, which can lead to proactive and persistent behavior [24]. In particular, a person’s psychological satisfaction in competence and autonomy plays an important role in one’s behavior [26]. Specifically, competence is described as an individual’s striving or need to experience a sense of accomplishment, achievement, or success. Autonomy refers to the need for individuals to decide their own behavior and engage in activities of their own choice [24]. In an Internet-mediated context, the opportunity for sharing knowledge is highly related to one’s competence perception, and the ability for self-presentation is often associated with autonomy perception [13,27].

For healthcare professionals, the psychological reward is a key consideration in motivating physicians to provide healthcare services online [12,17]. It is suggested that intrinsic motivations play an important role in physicians’ decisions regarding healthcare service provision. Optimizing physicians’ intrinsic motivations is beneficial to the healthcare industry at large in terms of healthcare expenditures and patients’ health [10]. In addition, extrinsic motivation can also influence healthcare professionals’ service behaviors online [28]. It is indeed found that economic benefits play an important role in motivating physicians to provide effective healthcare services [2,10]. Thus, it is hypothesized that:

**Hypothesis** **1:**
*Physicians who have a higher level of competence satisfaction have higher totals of service provision on healthcare service platforms.*


**Hypothesis** **2:**
*Physicians who have a higher level of autonomy satisfaction have higher totals of service provision on healthcare service platforms.*


**Hypothesis** **3:**
*Physicians who wish to gain higher economic benefits have higher totals of service provision on healthcare service platforms.*


### 2.4. The Moderating Effects of Offline Status

By being different from traditional online communities, a physician’s identity in an OHC is tied to an offline hospital or clinic [2]. A physician’s offline status is fundamentally a sociological concept that captures his/her social ranking in providing offline healthcare services, which can be measured based on their capabilities and performance in providing healthcare services [29,30,31]. In China, physicians have four medical titles (i.e., chief physician, deputy chief physician, resident physician, and attending physician) [32]. A physician’s medical title is considered to be representative of his/her offline status [2,33], which reflects a physician’s professional ranking and medical capacities.

Physicians with different offline statuses might behave differently even towards the same incentives [34]. For example, physicians with higher offline status are likely to pay more attention to achievements and self-fulfillments [35,36]. The offline status of physicians is likely to affect the impacts of the effect of competence satisfaction, autonomy satisfaction, and economic benefit on their healthcare service provision. Particularly, with a higher offline status, a physician’s competence would be more recognized offline with less time to provide healthcare service online. In addition, monetary reward is reported to have a stronger positive effect on physicians with low offline status than on physicians with high offline status [2]. Thus, the following hypotheses are proposed:

**Hypothesis** **4(1):**
*A physician’s offline status negatively moderates the relationship between competence satisfaction and service provision.*


**Hypothesis** **4(2):**
*A physician’s offline status negatively moderates the relationship between autonomy satisfaction and service provision.*


**Hypothesis** **4(3):**
*A physician’s offline status negatively moderates the relationship between economic benefit and service provision.*


Thus, a research model is proposed to depict the motivation of healthcare service provision and the moderating effects of offline status, as shown in Figure 1.

## 3. Methodology

### 3.1. Data Collection

This study used data from physicians specialized in coronary heart diseases from Haodf.com (http://www.haodf.com/, accessed on 1 March 2019). Haodf.com was founded in 2006 and is regarded as one of the most popular healthcare platforms in China, with currently 195,000 physicians providing healthcare services via the platform. Haodf.com is a viable platform for studying issues related to online healthcare services by allowing researchers to crawl data on the platform [37]. In this study, all physicians specialized in coronary heart diseases on the Haodf.com platform, and their interactions with patients were automatically downloaded using a crawler twice, in April 2019 and May 2019, respectively. After deleting invalid data, 2702 physicians’ data were retained for further analyses.

This study uses the number of patients as a proxy for a physician’s online healthcare service provision. In order to reduce reciprocal causality between dependent variables and independent variables, this study uses the difference between two periods of data to measure the dependent variable.

On Haodf.com, patients can give ratings about a physician’s service quality after having completed their consultation, which can serve as gratification of the physician’s need for competence. Ratings on a physician from the patients reflect the physician’s comprehensive capability for delivering healthcare service [38], which can be used to measure physicians’ competence satisfaction. Physicians can write and post articles related to healthcare on the platform, which can represent the level of physicians’ voluntary participation that stems from internal motivation within the online community. When physicians feel a sense of autonomy and satisfaction, they tend to write and post articles [39]. The gifts a physician receives from patients after providing healthcare service represent the economic benefit that the physician can gain on Haodf.com. Digital gifts are purchased by patients on the online platform as a kind of gratitude to the physician, which forms a part of a physician’s financial income and has been shown to influence physicians’ participation and contribution on online healthcare service platforms [40,41].

On the Haodf.com platform, there are four types of offline medical titles: director physician, associate director physician, chief physician, and physician. They are expressed as 4, 3, 2, and 1, respectively [11]. A physician’s offline title is presented on his/her homepage on the platform. Table 1 lists the variables and their descriptions.

It should be noted that, in China, hospitals are managed using a 3-level system [42]. Level 1 hospitals include community hospitals and healthcare centers that directly provide prevention, medical treatment, health care, and rehabilitation services to a certain community. Level 2 hospitals are regional hospitals that provide comprehensive medical and health services to multiple communities within the region and undertake limited teaching and scientific research tasks. Level 3 hospitals are tertiary hospitals that provide high-level specialized medical and health services to multiple regions and perform tertiary medical education and scientific research tasks. Usually, patients tend to go to higher-level hospitals to see a physician, which results in physicians in tertiary hospitals spending much time providing offline healthcare services. Therefore, we control the effect of the hospital level on physicians’ online service provision in this study. The descriptive statistics of the variables are presented in Table 2.

### 3.2. Empirical Model

The dependent variable, online consultation, is the total number of times of consultations provided by a physician on a healthcare service platform. All the dependent variables are non-negative integers, where count models are appropriate for analysis [43]. Poison regression models and negative binomial regression models are two types of popular count models. While the former applies when the conditional mean is equal to the distribution, the latter does not assume an equal mean and variance. In addition, negative binomial regression models introduce a parameter to correct for over-dispersion when the variance is greater than the conditional mean [44]. As shown in Table 2, the mean and variance of the dependent variables are quite different. Therefore, this study adopts the negative binomial regression model to explore physicians’ online consultation services.
(1)$$Pr\left(Y=y|\lambda ,\theta \right)=\frac{\Gamma \left(y+\theta \right)}{\Gamma \left(y+1\right)\Gamma \left(\theta \right)}{\left(\frac{\theta }{\theta +\lambda }\right)}^{\theta }{\left(\frac{\lambda }{\theta +\lambda }\right)}^{y}$$

The negative binomial distribution has two parameters: *θ* and *λ*. Parameter *θ* captures over-dispersion in the data. When *θ* = 0, the negative distribution is the same as the Poisson distribution. Parameter *λ* is the expected value of the distribution. This study takes the logarithmic transformation of network scale because the distributions of this variable are highly skewed and because using logarithmic transformation to scale this kind of variable is appropriate [1]. The negative binomial regression model with fixed effects is explicitly expressed as follows:
(2)$$\begin{array}{l} \text{Consultation}\;\text{service}\\ \;=\beta_{0}+\beta_{1}\text{Hospital}\;\text{level}\;1_{i}+\beta_{2}\text{Hospital}\;\text{level}\;2_{i}+\beta_{3}\text{Hospital}\;\text{level}\;3_{i}+\beta_{4}\text{In}({\text{Visits}}_{i}+1)\\ \;+\beta_{5}\text{Competence}\;{\text{satisfaction}}_{i}+\beta_{6}\text{Autonomy}\;{\text{satisfaction}}_{i}+\beta_{7}\text{Economic}\;{\text{benefit}}_{i}\\ \;+\beta_{8}\text{Offline}\;{\text{status}}_{i}+\beta_{9}\text{Competence}\;{\text{satisfaction}}_{i}*\text{Offline}\;{\text{status}}_{i}\\ \;+\beta_{10}\text{Autonomy}\;{\text{satisfaction}}_{i}*\text{Offline}\;{\text{status}}_{i}+\beta_{5}\text{Economic}\;{\text{benefit}}_{i}*\text{Offline}\;{\text{status}}_{i}+\varepsilon \end{array}$$
where $\beta_{0}-\beta_{11}$ are regression coefficients of covariates, and ε is the error terms with $\varepsilon \;i.i.d.N\left(0,\theta_{\varepsilon }^{2}\right)$.

## 4. Results

### 4.1. Correlations

This study estimates the models using STATA 15.0 software. The correlations of the variables are presented in Table 3, which reveals that the independent variables are correlated with the dependent variables. There are some correlations that are relatively high, which implies the possibility of multicollinearity. Tests of the VIFs of the variables show that all the VIFs are below the threshold of 2, suggesting that multicollinearity is not a serious issue in the dataset. The correlations between independent variables and the moderator variable are low, which has helped to derive stable results.

### 4.2. Empirical Results

The empirical results for testing the negative binominal regression model are shown in Table A1 in the Appendix A. Similar to the linear regression model, the effect of the independent variable on the dependent variable is determined by the regression coefficient. The positive (negative) sign of the regression coefficient indicates that the independent variable has a positive (negative) impact on the dependent variable. Table A1 shows that with only control variables in Model 1, and the independent variables, moderator and interaction terms are added in Model 2–6, respectively. The Log pseudolikelihood, Wald chi^2^, Prob > chi^2^, and Pseudo R^2^ are reasonable and statistically significant.

Hypothesis 1 and 2 posit that physicians who have a higher level of competence satisfaction and autonomy satisfaction have higher totals of service provision on healthcare platforms. According to the results of testing Model 2 shown in Table A1, the coefficient of competence satisfaction (*B* = 0.958, *p* < 0.01) is positive and statistically significant, which supports H1. The coefficient of autonomy satisfaction (*B* = 0.275, *p* < 0.01) is positive and statistically significant, which supports H2.

Hypothesis 3 proposes that physicians who wish to gain higher economic benefit have higher totals of service provision on healthcare platforms. According to the results of testing Model 3 shown in Table A1, the coefficient of economic benefit (*B* = 0.681, *p* < 0.01) is positive and statistically significant, which supports H3.

Hypothesis 4(1)–4(2) posit that a physician’s offline status negatively moderates the relationship between competence satisfaction, autonomy satisfaction, and consultation service. According to the results of testing Model 4 shown in Table A1, the coefficient of competence satisfaction ∗ offline status (*B* = −0.133, *p* < 0.01) is negative and statistically significant, which supports H4(1). However, the coefficient of autonomy satisfaction ∗ offline status (*B* = 0.053) is positive, which means that H4(2) is not supported.

Hypothesis 4(3) posits that a physician’s higher offline status moderates the effect of economic benefit and consultation service. According to the results of testing Model 5 shown in Table A1, the coefficient of economic benefit ∗ offline status (*B* = −0.045) is negative but statistically insignificant, suggesting that H4(3) is not supported.

### 4.3. Robustness Check

In order to examine the robustness of the results, this study collected data for another month, which is June 2019. After deleting invalid data, 2907 physicians’ data were obtained to test equation (2). The results are presented in Table A2 in the Appendix A, which is consistent with the results shown in Table A1. The results of the robustness check confirm that Hypotheses 1–4(1) are supported, while hypotheses 4(2) and 4(3) are not supported.

As shown in Table 4, 4 of 6, the hypotheses are supported.

## 5. Discussion and Conclusions

Based on the uses and gratifications theory and self-determination theory, this study reveals that intrinsic and extrinsic motivations play important roles in a physician’s healthcare service provision on healthcare service platforms. Negative binomial regression was employed to test the proposed hypotheses using empirical data collected from a popular Chinese healthcare service platform.

The results indicate that competence satisfaction, autonomy satisfaction, and economic benefit have positive impacts on a physician’s service provision behavior. The findings are consistent with existing research in that one’s behavior can be affected by both internal and external motivators [25], including physicians’ engagement in online healthcare service provision [17]. Especially economic gain plays an important role in explaining physicians’ such behavior [1,11], which is also true for the physician providing services offline [10]. In addition, this study reveals that, when having a higher level of offline status, a physician who has a higher level of competence satisfaction would be less likely to provide consultation services online. This is in alignment with prior studies that offline status is a moderator for physician contribution behaviors in online healthcare services [2,16,34].

This study contributes to the existing literature by integrating intrinsic and extrinsic motivations to investigate how they affect physicians’ healthcare service provision behavior online. Findings from this study may derive recommendations for improving the features and design of healthcare service platforms. For example, the platform should improve its search and navigation approach to provide patients with easy access to a physician’s webpage and establish guidance for physicians to realize the effectiveness of service pricing mechanisms.

This study has several limitations and future research directions. First, this study used cross-sectional analysis, which is unable to display the dynamics of physicians’ service provision behavior. Future research can adopt longitudinal data to observe the changes over time. Second, this study used one healthcare service platform, and physicians specialized in cardiology as the research context, which is a lack of evidence of the generalizability of the findings. Future research may consider testing the model in other similar contexts of the online healthcare service platform, and physicians specialized in other diseases to validate the findings. Third, a quality study may be conducted to understand the results of this study to understand the mechanisms behind the behavior.

## Figures and Tables

**Figure 1 ijerph-19-15135-f001:**
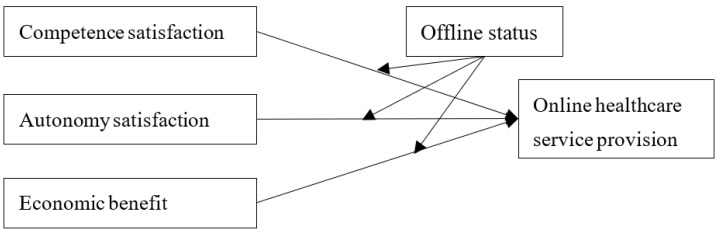
Research model.

**Table 1 ijerph-19-15135-t001:** Variable description.

Variable		Description	Proxy
Dependent variable	Online healthcare service provision	Total number of times of a physician providing online healthcare service	The total number of patients on the platform
Independent variables	Competence satisfaction	The level of a physician’s patients’ satisfaction with the physician’s online healthcare service	A physician’s online rating
	Autonomy satisfaction	The level of a physician’s voluntary contribution to the platform other than providing online healthcare service	The total number of articles published on the platform
	Economic benefit	The monetary returns a physician obtained online consultation	The total number of gifts received from patients
Moderator	Offline status	A physician’s offline social standing, which reflects his/her professional ranking and medical capacities	A physician’s offline medical title
Control variables	Hospital level	The level of the hospital where a physician registered with and working for offline	One of the three levels of hospital
	Visits	The level of attention from potential patients on the platform	The total number of visits of a physician’s homepage on the platform

**Table 2 ijerph-19-15135-t002:** Descriptive analysis of variables.

Variable	Min	Max	Mean	Std. Dev.
Hospital level 1	0	1	0.004	0.064
Hospital level 2	0	1	0.058	0.233
Hospital level 3	0	1	0.921	0.270
Visits	17	5.32 × 10^7^	448,430.200	1,625,244
Consultation service	0	389	8.137	26.282
Competence satisfaction	0	5	0.402	0.922
Autonomy satisfaction	0	1351	11.733	57.597
Economic benefit	0	4145	43.411	165.841
Offline status	0	4	3.038	0.912

**Table 3 ijerph-19-15135-t003:** Correlations of variables (*n* = 2702).

Variable	1	2	3	4	5	6	7	8
Hospital level 1	1							
Hospital level 2	−0.016	1						
Hospital level 3	−0.218 **	−0.842 **	1					
ln(Visits + 1)	−0.006	−0.039 *	0.026	1				
Competence satisfaction	−0.009	−0.073 **	0.084 **	0.193 **	1			
Autonomy satisfaction	0.028	0.002	−0.019	0.257 **	0.109 **	1		
Economic benefit	−0.013	−0.053 **	0.048 *	0.634 **	0.368 **	0.143 **	1	
Offline status	−0.054 **	−0.127 **	0.125 **	0.136 **	0.034	0.073 **	0.109 **	1

Note: ** Correlation is significant at the 0.01 level (2-tailed); * Correlation is significant at the 0.05 level (2-tailed).

**Table 4 ijerph-19-15135-t004:** Summary of results.

Hypothesis	Results	Tested in
Hypothesis 1: Physicians who have a higher level of competence satisfaction has higher totals of service provision on healthcare service platforms.	Supported	Model 2
Hypothesis 2: Physicians who have a higher level of autonomy satisfaction has higher totals of service provision on healthcare service platforms.	Supported	Model 2
Hypothesis 3: Physicians who wish to gain higher economic benefits has higher totals of service provision on healthcare service platforms.	Supported	Model 3
Hypothesis 4(1): A physician’s offline status negatively moderates the relationship between competence satisfaction and service provision.	Supported	Model 4
Hypothesis 4(2): A physician’s offline status negatively moderates the relationship between autonomy satisfaction and service provision.	Not supported	Model 4
Hypothesis 4(3): A physician’s offline status negatively moderates the relationship between economic benefit and service provision.	Not supported	Model 5

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
