# Peer review of "Understanding Physicians’ Motivation to Provide Healthcare Service Online in the Digital Age"

_ijerph, 2022, doi:10.3390/ijerph192215135_

Round 1
Reviewer 1 Report
This paper is well written in general. I think this topic is important and very interesting to readers. The authors studied factors that affect physicians’ healthcare service from online platform.
Positive factors and negative factors were concluded. The findings could be benifical for product design of online health service. Optimizing physicians’ intrinsic motivations is beneficial to the healthcare industry.
(1)Is the data used in this research public available? How can the interested readers reproduce the reported data modeling results?
(2)The language can be further improved.
(3)More recent references within three years, could be added to the current literature review.
(4) section 3.2 provides the empirical model as the main methodology part. Further discussion and comparison of this model against other existing methods is necessary.
Author Response
This paper is well written in general. I think this topic is important and very interesting to readers. The authors studied factors that affect physicians’ healthcare service from online platform.
Positive factors and negative factors were concluded. The findings could be benifical for product design of online health service. Optimizing physicians’ intrinsic motivations is beneficial to the healthcare industry.
Response: Thank you for your recognition and encouragement. With the helpful input that we have been given, we did our best to respond to and build upon the suggestions and ideas of the review team. In the revised manuscript, revisions are highlighted in red. We hope that you will be satisfied with our revisions.
(1)Is the data used in this research public available? How can the interested readers reproduce the reported data modeling results?
Response: We are afraid the data used in this research is not publicly available. The interested readers can contact the authors to obtain the dataset used in this study.
(2)The language can be further improved.
Response: Thank you for the suggestion. The language is further improved.
(3)More recent references within three years, could be added to the current literature review.
Response: Thank you for the suggestion. Eight recent references have been added, which are highlighted in the references.
(4) section 3.2 provides the empirical model as the main methodology part. Further discussion and comparison of this model against other existing methods is necessary.
Response: Thank you for letting us clarifying the reasons for choosing the model used in this study. In our study, because the data for measuring the dependent variable are non-negative integers, the count model is suitable and adopted. There are two types of count model, i.e., the negative binomial regression model and the poison regression model. In addition, because the conditional mean of the dependent variable is unequal to the distribution, the negative binomial regression model is suitable and adopted. (lines 219-226)
Reviewer 2 Report
Dear Authors,
Thanks for giving me the chance to read this manuscript, “Understanding physicians’ motivation to provide healthcare service online in the digital age”. The current paper tries to investigate the factors that affect physicians’ healthcare service provision behavior on healthcare service platforms.
This is an interesting topic in the field of health service provision. However, there are major issues in the current manuscript that should be carefully addressed to be further considered.
1. Method
· Why do models 1-6 not have one which contains competence, autonomy, and status?
· Why was H2 supported? Models 4 and 6 indicate that autonomy did not have a significant effect on the DV.
· How do you scrawl the data from haodf.com via python? Any references?
2. Discussion
· I did not see the in-depth discussion and conversation with the prior literature since the discussion part is, surprisingly, missing.
3. Language and format
· Many language issues are found in the manuscript. Please carefully proofread it.
To sum up, I personally like this paper. However, the problems should be addressed in order to be further considered. Hope these suggestions help.
Author Response
Dear Authors,
Thanks for giving me the chance to read this manuscript, “Understanding physicians’ motivation to provide healthcare service online in the digital age”. The current paper tries to investigate the factors that affect physicians’ healthcare service provision behavior on healthcare service platforms.
This is an interesting topic in the field of health service provision. However, there are major issues in the current manuscript that should be carefully addressed to be further considered.
Response: Thank you for your recognition and the constructive comments. With the helpful input that we have been given, we did our best to respond to and build upon the suggestions and ideas of the review team. In the revised manuscript, revisions are highlighted in red. We hope that you will be satisfied with our revisions.
- Method
- Why do models 1-6 not have one which contains competence, autonomy, and status?
Responses: Thank you for this comment. Among models 1-6, model 4 contains competence, autonomy, and status.
- Why was H2 supported? Models 4 and 6 indicate that autonomy did not have a significant effect on the DV.
Responses: In this study, we propose Hypothesis 2: Physicians who have a higher level of autonomy satisfaction has higher totals of service provision on healthcare service platforms, as shown in Section 2.3. We used stepwise regression to test the hypothesis. In Model 2, only control variables, competence and autonomy, are added to test the main effect of competence and autonomy on dependent variables. The results in Model 2 show that the coefficient of autonomy satisfaction (B=0.275, P<0.01) is positive and statistically significant. Therefore, autonomy satisfaction positively influences online healthcare service provision, which supports H2. However, Model 4 tests the moderating effect of the moderating variable, that is, the moderating effect of status on the influences of competence and autonomy on online healthcare service provision, and Model 6 is full equation.
We add Table 4 to clearly show a summary of results of hypothesis testing. (Line 288-289)
- How do you scrawl the data from haodf.com via python? Any references?
Response: Thank you for posing this question. Haodf.com is a viable platform for studying issues related to online healthcare service by allowing researchers to crawl data on the platform (lines 176-178). So, we developed a Python-based crawler program to automatically download physicians’ homepage information.
- Discussion
- I did not see the in-depth discussion and conversation with the prior literature since the discussion part is, surprisingly, missing.
Response: Thank you for pointing out this issue. In this revision, the results are discussed in relation to the prior literature. “The results indicate that competence satisfaction, autonomy satisfaction, and economic benefit have positive impacts on a physician’s service provision behavior. The findings are consistent with existing research in that ones’ behavior can be affected by both internal and external motivators [25], including physicians’ engagement in online healthcare service provision [17]. Especially, economic gain plays an important role in explaining physicians’ such behavior [1,11], which is also true for physician providing services offline [10]. In addition, this study reveals that, when having a higher level of offline status, a physician who has a higher level of competence satisfaction would less likely to provide consultation service online. This is in align with prior studies that offline status is a moderator for physician contribution behaviors in online healthcare services [2,16,34].” (Lines 297-306).
- Language and format
- Many language issues are found in the manuscript. Please carefully proofread it.
Response: Thank you for pointing out this issue. The revised manuscript is carefully proofread.
To sum up, I personally like this paper. However, the problems should be addressed in order to be further considered. Hope these suggestions help.
Response: These suggestions are very helpful. Thank you so much.
Reviewer 3 Report
Zhang et al investigates the factors that affect physicians’ healthcare service provision behavior on healthcare service platforms. It is a well structured manuscript with a very descriptive background and systematic hypothesis generation. This study employs sound statistics.
Some considerations
1. Include the content of lines 58-60m in the abstract to improve clarity
2. The surrogates used (ie posting publications) to measure different satisfaction domains are assumed to represent that specific domains. These are big assumptions. This need to be clarified for each of the outcomes measures, why you think this may be correlated. Explain the construct of receiving gifts as this is not a common practice in many fee for service healthcare systems.
3. In table 2 explain what Hospital level 1,2,3 are.
4. Line 206 should be MEAN. Please proof read throughout
5. Suggest including some of the statistics in to a supplementary appendix and leaving the ones that shows correlation in the main draft. The results section is very dense for the reader at present.
6. Have you addressed the model being too parsimonious with potential for overfitting. Have you done validation with ie boot strapping to assess the validity of your results
7. Under limitations discuss how this is derived from a specific clinic (cardiology it seems). This may limit generalizability to other groups and patient groups.
Author Response
Zhang et al investigates the factors that affect physicians’ healthcare service provision behavior on healthcare service platforms. It is a well structured manuscript with a very descriptive background and systematic hypothesis generation. This study employs sound statistics.
Some considerations
Response: Thank you for your recognition and the constructive comments. With the helpful input that we have been given, we did our best to respond to and build upon the suggestions and ideas of the review team. In the revised manuscript, revisions are highlighted in red. We hope that you will be satisfied with our revisions.
- Include the content of lines 58-60m in the abstract to improve clarity
Response: Thank you for the suggestion. The clarity of the abstract is improved by stating “A research model is proposed based on related literature and uses and gratifications theory and self-determination theory. The empirical data were collected from a popular Chinese healthcare service platform, and negative binomial regression was employed to test the proposed research model.” (Lines 12-15)
2. The surrogates used (ie posting publications) to measure different satisfaction domains are assumed to represent that specific domains. These are big assumptions. This need to be clarified for each of the outcomes measures, why you think this may be correlated. Explain the construct of receiving gifts as this is not a common practice in many fee for service healthcare systems.
Response: Thank you for the suggestion. The measures are clarified in this revision: “On Haodf.com, patients can give ratings about a physician's service quality after having completed their consultation, which can serve as gratification of the physician’s need for competence. Ratings on a physician from the patients reflect the physician’s comprehensive capability for delivering healthcare service [38], which can be used to measure physicians’ competence satisfaction. Physicians can write and post articles related to healthcare on the platform, which can represent the level of physicians’ voluntary participation that stems from the internal motivation within the online community. When physicians feel a sense of autonomy satisfaction, they tend to write and post articles [39].The gifts a physician received from patients after providing healthcare service represents the economic benefit that the physician can gain on Haodf.com. Digital gifts are purchased by patients on the online platform as a kind of gratitude to the physician, which forms a part of a physician’s financial income and has been shown to influence physicians’ participation and contribution on online healthcare service platforms [40,41].” (Lines 186-198)
3. In table 2 explain what Hospital level 1,2,3 are.
Response: Thank you for giving the opportunity to clarify this. The Hospital level 1,2,3 are explained: “In China, hospitals are managed using a 3-level system [39]. Level 1 hospitals include community hospitals and healthcare centers that directly provide prevention, medical treatment, health care and rehabilitation services to a certain community. Level 2 hospitals are regional hospitals that provide comprehensive medical and health services to multiple communities within the region and undertake limited teaching and scientific research tasks. Level 3 hospitals are tertiary hospitals which provide high-level specialized medical and health services to multiple regions and performing tertiary medical education and scientific research tasks.” (Lines 205-212)
4. Line 206 should be MEAN. Please proof read throughout
Response: Thank you for the careful reading. The typo in line 206 is corrected (in line 226 in this revision). A throughout proofreading is conducted.
5. Suggest including some of the statistics in to a supplementary appendix and leaving the ones that shows correlation in the main draft. The results section is very dense for the reader at present.
Response: Thank you for the suggestion. The two tables, Table 4 and Table 5 in the original manuscript, reporting the results of parameter estimations are moved to the Appendix as Table A1 and Table A2.
6. Have you addressed the model being too parsimonious with potential for overfitting. Have you done validation with ie boot strapping to assess the validity of your results
Response: Thank you for the comments. Yes, we addressed the potential overfitting issue by proposing a relatively simple model with three independent variables, one moderator and two control variables (see Table 1 in line 203), using a large sample size of 2702 to examine the research model (line181), and collecting additional data to further validate the model (Lines 282-283). The validity of the results is assessed using a robustness check (see Section 4.3).
7. Under limitations discuss how this is derived from a specific clinic (cardiology it seems). This may limit generalizability to other groups and patient groups.
Response: Thank you for the suggestion. Discussion on this limitation is presented in this revision: “this study used one healthcare service platform and physicians specialized in cardiology as the research context, which is lack of evidence of generalizability of the findings. Future research may consider testing the model in other similar contexts of online healthcare service platform and physicians specialized in other diseases to validate the findings.” (lines 317-321)
Reviewer 4 Report
It is a good effort. Minor language editing needed.
It would be beneficial to provide effect sizes as well.
Author Response
It is a good effort. Minor language editing needed.
Response: Thank you for the positive comment and suggestion. The language is improved.
It would be beneficial to provide effect sizes as well.
Response: Thank you for the suggestion. In this study, we followed prior studies in reporting the results of data analysis without providing effect sizes [43]. But we will consider provide effect sizes in future studies.
Reviewer 5 Report
Review of the article "Understanding physicians’ motivation to provide healthcare service online in the digital age".
The article has been written carefully and the research presented has been done correctly. The subject covered in the article is important.
I have two comments to the article:
1. Lines 34-35, there is: "Thus, online healthcare service platforms can be considered as an alternative and complementary means to the traditional healthcare service system"
In my opinion, online healthcare service platforms can be considered as an alternative and complementary means to the traditional healthcare service system only in special cases. Nothing can replace the direct meeting of the patient with the doctor. In my opinion, if it is possible, the patient should meet the doctor directly. So, I agree with the sentence that is in the lines 70-72, i.e. "Some studies have found that, even though physicians provide services on the online healthcare service platform, they are less willing to take the initiative to learn about patients, resulting in mixed service quality".
As this conclusion is somewhat contradictory to the conclusions presented in this article, the Authors should discuss it in detail.
2. Immediately after reading the title of the article, I thought that a factor that I later noticed in lines 78-79, might dominate and distort the results and distort the conclusions.
Lines 78-79, there is: "However, other studies have found that financial rewards (such as service fees and gifts) they obtain online also promote physicians’ online service behavior")
So, the use of services on the online healthcare service platform may be dictated more by profit, not real help for the patients. The Authors should discuss this problem in detail.
Author Response
The article has been written carefully and the research presented has been done correctly. The subject covered in the article is important.
I have two comments to the article:
Response: Thank you for your recognition and the constructive comments. With the helpful input that we have been given, we did our best to respond to and build upon the suggestions and ideas of the review team. In the revised manuscript, revisions are highlighted in red. We hope that you will be satisfied with our revisions.
1. Lines 34-35, there is: "Thus, online healthcare service platforms can be considered as an alternative and complementary means to the traditional healthcare service system"
In my opinion, online healthcare service platforms can be considered as an alternative and complementary means to the traditional healthcare service system only in special cases. Nothing can replace the direct meeting of the patient with the doctor. In my opinion, if it is possible, the patient should meet the doctor directly. So, I agree with the sentence that is in the lines 70-72, i.e. "Some studies have found that, even though physicians provide services on the online healthcare service platform, they are less willing to take the initiative to learn about patients, resulting in mixed service quality".
As this conclusion is somewhat contradictory to the conclusions presented in this article, the Authors should discuss it in detail.
Response: Thank you for the pointing this issue. We explain some special cases where online healthcare platforms can be considered as an alternative and complementary means to the traditional healthcare service system by stating “On the one hand, online healthcare service platforms replaces the traditional counterpart for delivering healthcare services, such as chronic diseases [6]. On the other hand, online healthcare service platforms complement the traditional healthcare service system by providing some services remotely, such as following up after treatment offline and patient education [7].” (Lines 37-41) This study focuses on hypertension, one type of chronic diseases, which belongs to one of the special cases.
2. Immediately after reading the title of the article, I thought that a factor that I later noticed in lines 78-79, might dominate and distort the results and distort the conclusions.
Lines 78-79, there is: "However, other studies have found that financial rewards (such as service fees and gifts) they obtain online also promote physicians’ online service behavior")
So, the use of services on the online healthcare service platform may be dictated more by profit, not real help for the patients. The Authors should discuss this problem in detail.
Response: Thank you for the comment. Yes, profit indeed plays an important role in explaining physicians’ use of services on the online healthcare service platform. Even for physicians to provide service offline, profit is a key motivator (Lines 301-303). However, in addition to profit, helping for patients also plays a role in motivating physicians to provide service online (Lines 298-301).
Round 2
Reviewer 2 Report
The authors have addressed most of my concerns. I am happy to recommend it for publication.
Reviewer 5 Report
The article has been corrected properly and carefully, taking into account all my concerns and comments. Now, in my opinion, the article can be published in International Journal of Environmental Research and Public Health.